# Optimization of Extraction of Phenolic Compounds with Antimicrobial Properties from *Origanum vulgare*

**Taja Žitek** [1], **Dragana Borjan** [1], **Andrej Golle** [2], **Željko Knez** [1,3] and **Maša Knez** [1,*]

1 Laboratory for Separation Processes and Product Design, Faculty of Chemistry and Chemical Engineering, University of Maribor, SI-2000 Maribor, Slovenia; taja.zitek@um.si (T.Ž.); dragana.borjan@um.si (D.B.); zeljko.knez@um.si (Ž.K.)

2 National Laboratory for Health, Environment and Food, Prvomajska ulica 1, SI-2000 Maribor, Slovenia; andrej.golle@nlzoh.si

3 Faculty of Medicine, University of Maribor, Taborska 8, SI-2000 Maribor, Slovenia

* Correspondence: masa.knez@um.si; Tel.: +386-2-229-44-70

**Abstract:** Oregano (*Origanum vulgare*) is considered to be a good and cheap source of phenolic compounds with favorable biological activities, especially antimicrobial and antioxidant properties. Hypothesis/Purpose: The current work explored the optimization of the process conditions of solid–liquid extraction from *Origanum vulgare* to obtain extracts with high antimicrobial activity. We investigated which parameters promoted different efficiencies, leading to the maximum extraction of phenols and the consequent highest level of biological activity. Design-Expert Pro 11 was selected to design and analyze the experiments. The extracts were obtained by maceration as a simple method to recover value-added compounds from plant material, and supercritical fluid extraction was carried out as a green method with a high selectivity to obtain the compounds of interest. Pressure, temperature, and time were varied to obtain extracts with high antioxidant and antimicrobial activity. According to the results obtained using Design-Expert, the optimal conditions for maceration were at a temperature of 83 °C. The 1,1′-diphenyl-2-picrylhydrase method was used for the determination of antioxidant potential, while microdilution methods were used to determine the antimicrobial potential with regard to *Staphylococcus aureus*, *Escherichia coli*, and *Candida albicans*. A level of antioxidant activity of 87.21% was achieved. Supercritical fluid extracts showed higher antioxidant activity at a higher temperature of 60 °C and higher pressure of 25 MPa, although the results at 40 °C and 25 MPa were similar. The lowest minimum inhibitory concentration (MIC) values were 0.147 mg/mL for *S. aureus*, 0.728 mg/mL for *E. coli*, and 0.311 mg/mL for *C. albicans*. Overall, the optimal conditions for supercritical fluid extraction were 25 MPa and 40 °C. On the other hand, amounts of 0.208 mg/mL for *S. aureus*, 1.031 mg/mL for *E. coli* and 0.872 mg/mL for *C. albicans* were obtained using maceration. The MIC values of extracts obtained by supercritical fluid extraction were comparable to the minimum inhibitory concentration values obtained by different conventional techniques, such as those of Clevenger and Soxhlet.

**Keywords:** oregano extracts; supercritical fluid extraction; cold maceration; antioxidant activity; antimicrobial activity; *Staphylococcus aureus*; *Escherichia coli*; *Candida albicans*

## 1. Introduction

Oregano (*Origanum vulgare*) is a culinary and medicinal herb from the mint or *Lamiaceae family*. It typically grows to around 50 cm tall and has purple leaves that are around 2 to 3 cm in length [1]. The chemicals that give the herb its unique and pleasant smell are thymol, pinene, limonene, carvacrol, ocimene, and caryophyllene. It adds flavor, can inhibit the growth of microorganisms, and may have several health benefits [2–4].

Oregano contains polyphenolic compounds (acacetin, apigenin, diosmetin, herbacetin, hispidulin, luteolin, naringin, quercetin, and rosmarinic acid [5–7]) that have multiple biological effects [7–10], including antioxidant and antimicrobial activity [3,4]. Oregano is a

strong candidate for a natural food preservative because of its antioxidant and antimicrobial action, probably due to the presence of the diterpenes carvacrol and thymol, which have been extensively studied for their anti-bacterial and antioxidant properties [7,11].

Since the body is not capable of producing certain antioxidants on its own, they must be obtained through food; in this context, the preparation or treatment of the material is important to preserve the biologically active components. Maceration is a very common process in cooking, such as when used to make tea or when oregano is added to various sauces, especially in the preparation of Mediterranean dishes [12]. In the maceration process, the material is dissolved in a solvent and mixed for a specific period, until the material structure is softened and penetrated by the solvent and the soluble constituents are extracted. Justė Baranauskaitėa and others have noted the impact of the preparation of the extract on the resulting content of the active components in oregano, such as carvacrol, rosmarinic acid, urosolic acid, and oleoanolic acid [13]. Cervato et al. also reported high levels of antioxidants in aqueous oregano extract [14]. It has been reported that supercritical extracts are of superior quality, with greater biological activity compared to extracts produced with liquid solvents, and that there is less deterioration of the thermally labile components in the extract, with higher antioxidant contents than can be achieved with maceration. It is assumed that the polarity of the solvent used in the maceration process has significant effects on the amount of antioxidants in the resulting extracts and the total phenolic profile [15,16]. Water is a polar solvent, and it is suitable for the extraction of polar compounds such as phenols.

The use of so-called "green extraction techniques" and "green solvents" has gained increased interest over the past few years, with $CO_2$ being one of the greenest solvents conceivable, after water.

The aim of this study was to determine the optimal extraction conditions of oregano by applying the Design-Expert Pro 11 software, with time and temperature being the observed variables. Further, supercritical fluid extraction was carried out, and the effects of pressure were also examined. In the first step, the extracts were evaluated for their antioxidant activity to obtain a preliminary indication of their biological potential by using the 2,2-diphenyl-1-picrylhydrazyl (DPPH) method. Antimicrobial activity was determined for *Staphylococcus aureus*, *Escherichia coli*, and *Candida albicans.*

To evaluate the effects of the predicted model with regard to the response variables, an analysis of variance (ANOVA) with a 95% confidence level was carried out to assess the effect of each factor (temperature, time, and ratio). In addition, the regression coefficient (R2), the *p*-value of the regression model, and the *p*-value of the lack-of-fit (LOF) were used to determine the fitness of the regression model. Optimal conditions were chosen considering the response surfaces (3D plots).

The optimized conditions were validated for the antioxidant activities (DPPH) based on the values obtained using response surface methodology (RSM). All the responses were determined under optimized conditions of extraction. The experimental values were compared with the predicted values to determine the validity of the model.

Therefore, in the temperature range of 60 °C to 100 °C and with times of 10 min to 30 min, the optimal antioxidant point was determined for the maceration process and compared with the results for supercritical fluid extraction (SFE) for the temperature and pressure variables (40 °C to 60 °C and 15 MPa to 25 MPa). The antimicrobial potential of both leading extracts was measured, and the results were compared with those in the literature.

## 2. Materials and Methods

### 2.1. Materials

Wild oregano (*Origanum vulgare*) was purchased from Alfred Galke GmbH (Samtgemeinde, Bad Grund, Germany) and ground before use. The $CO_2$ used in the SFE process was obtained from Messer, MG-Ruše, Slovenia, with a purity of 2.5. For the determination of antioxidant activity, 2,2-diphenyl-1-picrylhydrazyl (Sigma Aldrich, Darmstadt,

Germany, ≥97.0%) and methanol (MeOH), (Honeywell, Charlotte, NC, USA, LC-MS CHROMASOLV®, ≥99.9%) were used. The oregano extract was suspended in Mueller–Hinton agar with the emulsifying agent Polysorbate 80-Tween 80 (T80). Antimicrobial activity was examined using three different microorganisms, namely *Staphylococcus aureus* (ATCC 25923), *Escherichia coli* (ATCC 25922) and *Candida albicans* (ATCC 60193), all added to Mueller-Hinton broth (MH) obtained by National Laboratory for Health, Environment and Food.

### 2.2. Methods

### 2.2.1. Design-Expert Pro 11

The software Design-Expert Pro 11 was selected to design and analyze the experiments. Design-Expert is a statistical software package from Stat-Ease Inc. (Minneapolis, Minnesota, United States) that is specifically designed to run Experimental Design (DOE) [17]. The program was used to produce a diagram to determine the optimal extraction conditions (temperature and time).

### 2.2.2. Maceration

The ground, dried oregano (2 g) and water (150 mL) were both added to a flask. In the experimental design, two operating parameters were varied: extraction time (min) and temperature (°C). The extraction temperatures were 60 °C, 80 °C, and 100 °C, while the times were 10 min, 20 min, and 30 min. After every completed extraction, the solvent was removed at 40 °C under reduced pressure using a rotary evaporator (Büchi Rotavapor R-114, Flawil, Switzerland). Extract was stored in the fridge at temperature −4 °C for less than one month. Extracts were used for further analysis. This method was evaluated in Design-Expert Pro 11.

### 2.2.3. Supercritical Fluid Extraction (SFE)

The experiment was performed using the SFE system, as explained in detail by Žitek et al. (2020) [18]. The ground, dried oregano (10 g) was placed into a high-pressure vessel. The extraction was performed at different conditions of pressure (15 MPa and 25 MPa) and temperature (40 °C and 60 °C). The feed of the solvent ($CO_2$) was F/S = 8.168. The scheme of the SFE apparatus is shown in Figure 1, which consisted of an autoclave, separator, and high-pressure (HP) pump as well as corresponding pipes and valves. The temperature was controlled by the heater and the gas was introduced through the pipes with the compressor. The extract was collected at the bottom of the separator into a test tube as a result of the reduction in pressure [19,20].

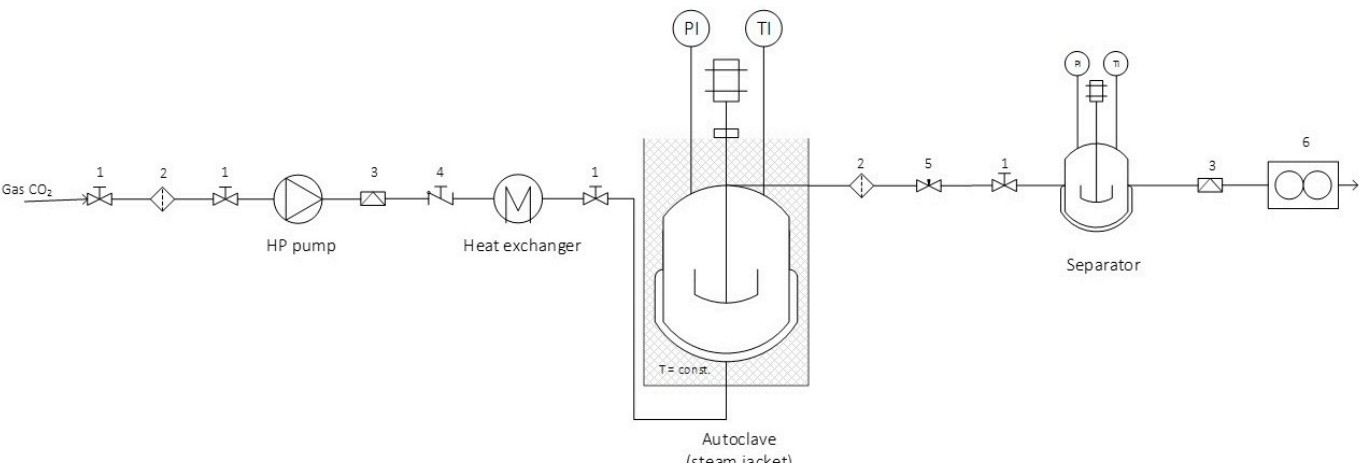

**Figure 1.** Supercritical fluid extraction system: 1—valve, 2—HP filter, 3—rupture disk, 4—one-way valve, 5—regulating valve, and 6—flowmeter [18].

2.2.4. Determination of Antioxidant Potential

Antioxidant activity was measured using the 1,1′-diphenyl-2-picrylhydrase (DPPH) method at a wavelength of 515 nm, as described in an earlier study [21]. The antioxidative activity of the sample is given as a percentage of inhibition relative to the reference solution and is calculated by the following equation, where $A_0$ represents the absorbance of the reference solution for 0 min and $A_{15}$ represents the absorbance of the solution for 15 min [21,22]:

$$A[\%] = \left( \frac{A_0 - A_{15}}{A_{15}} \right) \cdot 100\% \tag{1}$$

2.2.5. Procedure of Emulsification

Mueller–Hinton broth (MH) is water-based and the oregano extract is oil-based, and since these are immiscible, a process of emulsification was needed. Approximately 75 mg of oregano extract was measured and heated at 40 °C. In total, 100 μL of Tween 80 (T80) emulsifying agent was added at room temperature, while 900 μL of MH was homogenized at 40 °C. Oregano extract was suspended in MH using a rotor–stator homogenizer (Homogenizer, Polytron Pt1200, Kinematica AG, Luzern, Switzerland). The extract suspension in MH was homogenized at 70,000 g.

2.2.6. Determination of Antimicrobial Potential

The microdilution method was performed using cation-adjusted MH broth, and the MH broth was supplemented with lysed horse blood and β-NAD (MH-F broth). The antimicrobial potential of *Staphylococcus aureus* (MH), *Escherichia coli* (ATCC 25922, ATCC, Wesel, Germany) and *Candida albicans* (ATCC 60193, ATCC, Wesel, Germany) was determined using the method presented in Žitek et al. (2020) [18].

Twelve columns were used for 96 wells. Each well was filled with 100 μL of MH broth. Then, 100 μL of the prepared extract was added to the first well and mixed. The dilution process was continued as 100 μL from the first column was transferred to the second column and mixed. The procedure was repeated until the 10th column, from which 100 μL of the mixture was thrown away. Consequently, each column from the 1st to the 10th was composed of 100 μL extract and MH broth, where the concentration decreased by half for each column. The 11th and 12th columns were used for positive and negative controls, respectively. The positive control contained 50 μL of MH broth and 50 μL of diluted extract, and the negative control contained 100 μL of MH broth and 10 μL of bacterium. Column 11 contained 10 μL of bacterium (inoculum density $10^8$ CFU/mL) added from each of the first 10 columns and mixed together. Columns with no color change (i.e., the blue resazurin color remained unchanged) were scored as above the MIC value. All assays were performed in triplicate.

2.2.7. Validation of the Model

The optimal process conditions of extraction (time and temperature, along with pressure in the case of SFE) were examined for the maximum in vitro antioxidant activities (DPPH) on the basis of the obtained values using RSM. All the responses were defined under the optimized conditions of extraction. The experimental values were compared with those anticipated by the model to assess its validity.

**3. Results and Discussion**

*3.1. Determination of Antioxidant Activity by DPPH Method*

Design-Expert Pro 11 produced an experimental design, and based on the results, the program proposed the following quadratic equation for calculating antioxidant activity:

$$A = -13.92381 + 2.42917 \cdot T - 0.014587 \cdot T^2 \tag{2}$$

where A is antioxidant activity (%) and T is solvent temperature (°C).

According to the equation, time does not represent a significant factor in the extraction of oregano, because the program eliminated it. It is also evident that the interaction of temperature and time has no significance. The only component that has an obvious impact is temperature. The analysis of variance (ANOVA) showed the fit of our model (F = 19.63 or $p < 0.0500$), and the inadequacy due to model mismatch was also insignificant ($p = 0.1962$). The assumed Pearson's squared coefficient (pred. $R^2$) had a value of 0.6569, which was within acceptable limits compared to the recommended $R^2$ of 0.7564. A ratio of 8.7938 was obtained for the coefficient of measurement accuracy, which was appropriate since a coefficient greater than 4 was acceptable.

The graphs in Figures 2 and 3 show that the optimal point for achieving the highest antioxidant content in maceration is between 70 °C and 90 °C. As with Equation (2), the graphs show that a maceration time in the range of 10 to 30 min does not represent a significant variable.

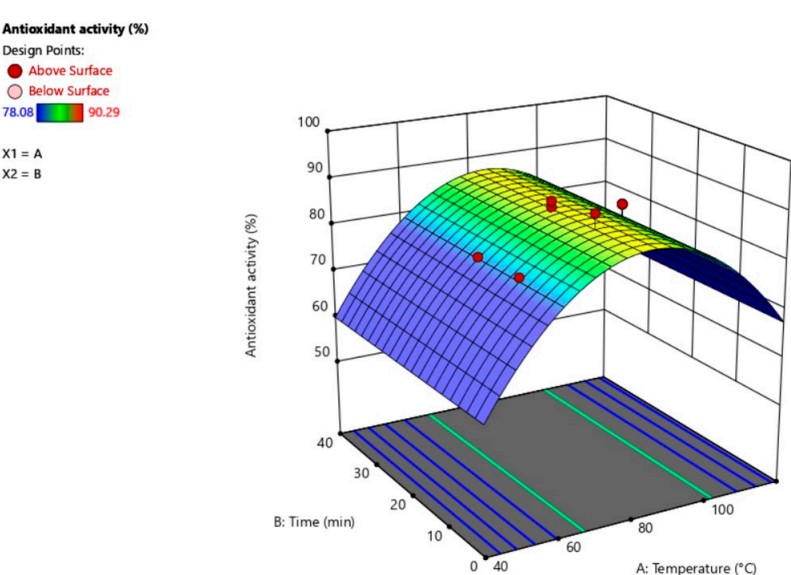

**Figure 2.** Three-dimensional response surface of antioxidant activity of oregano extract with respect to temperature and time.

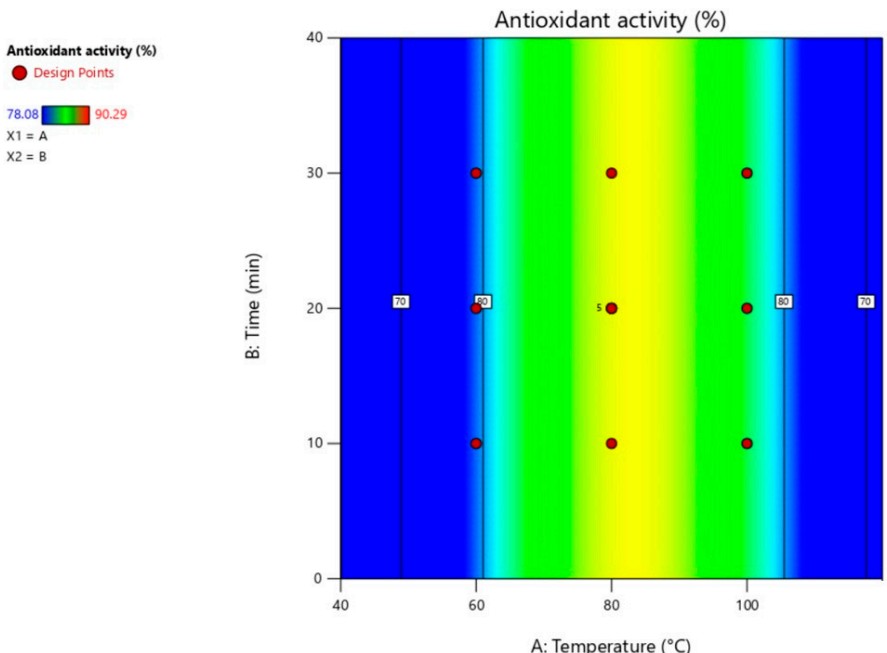

**Figure 3.** Two-dimensional contour diagram.

The results show that the optimal temperature with respect to antioxidant activity (A = 87.21%) is 83 °C. Oregano is one of the most used herbs in everyday cooking, so maceration with water as a solvent was chosen for comparison with SFE.

SFE was carried out at two different temperatures (40 °C and 60 °C) and two different pressures (15 MPa and 25 MPa), as can be seen in Figure 4. The highest antioxidant activity (A = 85.03%) was achieved at 25 MPa and 60 °C. The antioxidant content of the macerated extract is similar to that of SFE extract (25 MPa and 60 °C). Given the deviation error, the results are the same.

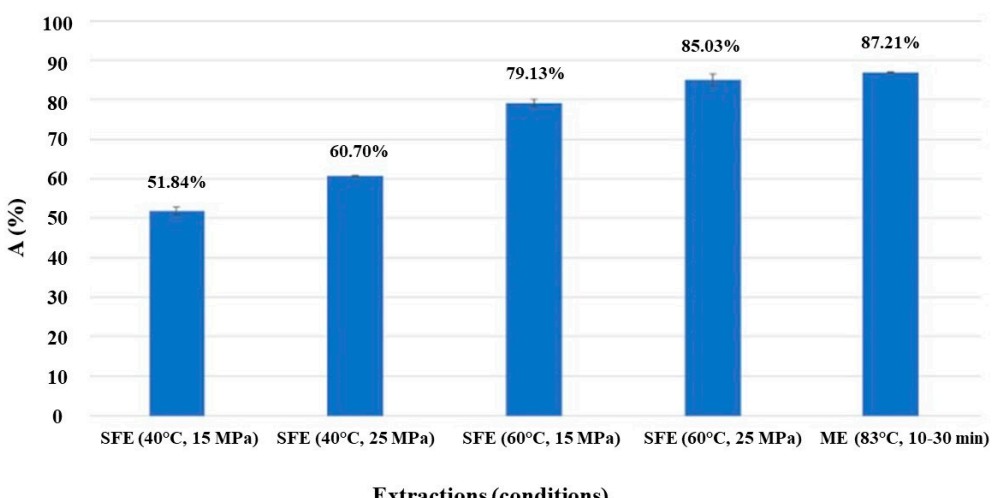

**Figure 4.** Results for the antioxidant activity of extracts obtained under different pressures and temperatures (SFE = supercritical fluid extraction and ME = maceration extraction).

At a temperature of 40 °C and pressures of 15 MPa and 25 MPa, antioxidant activities of 51.84% and 60.70% were obtained, respectively, whereas the values obtained at 60 °C under the same pressures were 79.13% and 85.03%, respectively. The results obtained by SFE at the lowest temperatures were similar to those obtained by maceration (the working temperature for SFE was 60 °C and for ME was 83 °C). In this case, the antioxidant activity of the supercritical fluid extract was 85.03% at a temperature of 60 °C and a pressure of 25 MPa, whereas the antioxidant activity in the case of maceration was 87.21% at a significantly higher temperature of 83 °C. From an industrial point of view, using less energy for heating can be an additional benefit. The results obtained with maceration are in line with those in the literature, where it was also found that oregano extracts prepared by maceration with hot water or with cold water showed more antioxidant activity with the former [22]. The results in the literature are also in line with those in this study for SFE. SFE with ethanol as a co-solvent gave better results than those found when using only $CO_2$, but it would be preferable to eliminate the application of an organic solvent in order to enable further research into antimicrobial activity, as ethanol could have a negative influence on microorganisms [23,24].

### 3.2. Minimum Inhibitory Concentration (MIC)

*E. coli* can be ingested with contaminated food, such as undercooked ground beef and fresh products. On the other hand, *S. aureus* can be found on the skin or in the nose of even healthy individuals, but it can turn deadly if the bacteria invade deeper into the human body. Additionally, the fungus *C. albicans* is part of the mouth's natural microflora, but it can infect skin and mucus membranes. The MIC values of five different oregano extracts against these two bacteria and one fungus, as obtained in this study, are shown in Table 1. This shows the average of all measurements, where the standard variation is between 0.003 and 0.007. It can be observed that all microorganisms were susceptible to the action of oregano extracts, with a variation in the MIC values from 0.147 to 2.712 mg/mL.

The oregano extract obtained under a pressure of 25 MPa and temperature of 40 °C had the most significant impact on each tested microorganism.

**Table 1.** Determination of the MIC by the resazurin dye microdilution method for five different extracts against two standard bacteria and one fungus.

| Microorganism | Extractions | Conditions | MIC (mg/mL) |
|---|---|---|---|
| *Staphylococcus aureus* ATCC 25923 gram-positive bacterium | SFE | 40 °C, 25 MPa | 0.277 |
| | SFE | 40 °C, 15 MPa | 0.293 |
| | SFE | 60 °C, 25 MPa | 0.147 |
| | SFE | 60 °C, 15 MPa | 0.327 |
| | ME | 83 °C, 10–30 min | 0.208 |
| *Escherichia coli* ATCC 25922 gram-negative bacterium | SFE | 40 °C, 25 MPa | 2.484 |
| | SFE | 40 °C, 15 MPa | 1.356 |
| | SFE | 60 °C, 25 MPa | 1.211 |
| | SFE | 60 °C, 15 MPa | 0.728 |
| | ME | 83 °C, 10–30 min | 1.031 |
| *Candida albicans* ATCC 60193 fungus | SFE | 40 °C, 25 MPa | 0.311 |
| | SFE | 40 °C, 15 MPa | 2.712 |
| | SFE | 60 °C, 25 MPa | 2.421 |
| | SFE | 60 °C, 15 MPa | 1.457 |
| | ME | 83 °C, 10–30 min | 0.872 |

ATCC—American Type Culture Collection (USA).

The MIC values obtained by SFE and maceration in our study were comparable to those in the literature. Busatta et al. (2007) reported the antimicrobial activity of oregano essential oil against several bacteria, including *S. aureus* and *E. coli*. The extractions were performed in a Clevenger apparatus. The MIC results obtained in their work were 0.230 mg/mL in the case of *S. aureus* and 0.460 mg/mL in that of *E. coli* [25]. Compared to these results, the MIC values determined in our work were comparable for both bacteria, ranging from 0.147 to 0.327 mg/mL for *S. aureus* and from 0.728 to 2.484 mg/mL for *E. coli*. Sarikurkcu et al. (2015) researched the antimicrobial potential of oregano essential oil distilled using a Clevenger apparatus. They also used the agar well diffusion method with MH broth, but with a lower inoculum density of microorganisms ($10^6$ CFU/mL) and volume in 100 µL of MH broth (0.5 µL). The reported MIC values were 0.107 mg/mL for *S. aureus*, 0.213 mg/mL for *E. coli,* and 0.128 mg/mL for *C. albicans* [26]. Although we used a higher inoculum density of microorganisms, our results were comparable with those in the literature, with even better MIC values. Kosakowska et al. (2021) reported a study into oregano essential oils and extracts of two different species (Greek oregano and common oregano) prepared by the Soxhlet extraction method. They obtained MIC values from 4 mg/mL to 32 mg/mL for *S. aureus* and from 4 mg/mL to 64 mg/mL for *E. coli*, which are higher than the values obtained in this study [27]. Ličina et al. (2013) studied the antimicrobial activity of oregano extracts obtained using different solvents (water, ethanol, acetone, ethyl-acetate, and diethyl ether). They reported MIC values against *E. coli* ATCC 25922 from 2.50 mg/mL to 20.0 mg/mL, against *S. aureus* ATCC 25923 from 0.160 mg/mL to 0.60 mg/mL, and against *C. albicans* from 10.0 mg/mL to 62.5 mg/mL [28]. Against *E. coli,* all five extracts used in our work gave comparable results. Our results against *S. aureus* were similar to those mentioned above. Furthermore, the MIC values against *C. albicans* obtained in our work using five different oregano extracts were significantly lower, ranging from 0.311 mg/mL to 2.712 mg/mL. De Martino et al. (2009) studied the influence of oregano from three different locations on two of the same bacterial species as presented in this work (*S. aureus* ATCC 25,923 and *E. coli* ATCC 25922). They used a Clevenger apparatus for isolation, and for the antimicrobial tests, the MIC method was employed. MIC values ranging from 0.050 to 0.100 mg/mL were reported for both bacteria [29]. Boskovic et al. (2015) investigated the antimicrobial effects of oregano on some food-borne bacteria, such as *E. coli* ATCC 25922 and *S. aureus* ATCC 25923. MICs were determined by the broth

microdilution method, and values obtained against *E. coli* and *S. aureus* were 0.320 mg/mL and 0.640 mg/mL, respectively [30]. Our results against *E. coli* were comparable with these findings. The results obtained against *S. aureus* were approximately half those in the earlier study. The antibacterial activity of oregano essential oil on *E. coli, C. albicans,* and *S. aureus* was determined by Özkalp et al. (2010). They obtained MICs of 0.250 mg/mL against *E. coli*, 0.064 mg/mL against *C. albicans*, and 0.064 mg/mL against *S. aureus* [31].

## 4. Conclusions

During this research, the optimal maceration temperature and time were determined using Design-Expert Pro 11. Supercritical fluid extraction was also carried out to compare the antioxidant activity, since the implementation required a much lower temperature, which would significantly reduce the costs of production at higher volumes. In terms of antioxidant activity and antimicrobial activity, the extracts obtained by maceration at 83 °C (A = 87.21%) and supercritical $CO_2$ extracts at 25 MPa pressure and 60 °C (A = 85.03%) could be compared. The maceration results revealed that extending the time to 30 min had no significant effect on the antioxidant activity of the extract, while the variation of the temperature had a notable impact. Higher antioxidant activities were measured with the SFE extracts at higher temperatures and pressures. The oregano extract obtained at a pressure of 25 MPa and temperature of 40 °C had the most significant impact on each of the tested microorganisms, and the results were also comparable with those of the macerations. The results obtained in this work are comparable to the MIC values obtained by different conventional techniques, such as those of Clevenger and Soxhlet.

**Author Contributions:** M.K. prepared the concept of this work. T.Ž. performed most of the experimental work and, with D.B., performed an extensive literature search. T.Ž. and M.K. wrote most of the paper. A.G., M.K. and Ž.K. devised the content of the review and supervised the writing. Financial management of project was conducted by Ž.K. All authors have read and agreed to the published version of the manuscript.

**Funding:** Financial support from the Slovenian Research Agency through grant P2-0046 and Smart Materials for Bio Applications J2-1725 is gratefully acknowledged.

**Institutional Review Board Statement:** Not applicable.

**Informed Consent Statement:** Not applicable.

**Data Availability Statement:** Not applicable.

**Conflicts of Interest:** There are no conflicts of interest.

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
