# Peer review of "Optimization of Extraction of Phenolic Compounds with Antimicrobial Properties from Origanum vulgare"

_processes, doi:10.3390/pr9061032_

Round 1
Reviewer 1 Report
Title: Optimization of extraction of phenolic compounds with antimicrobial properties from Origanum Vulgare
This manuscript describes to determine the optimal extraction conditions by applying the Design-Expert pro 11 software, with time and temperature being the observed variables. Further, supercritical fluid extract was carried out, and the effects of pressure were also examined. In the first step, the extracts were evaluated for their antioxidant activity for a preliminary indication of their biological potential by using the DPPH method. Higher antioxidant activities were found with the supercritical fluid extraction (SFE) extracts at higher temperatures and pressures. Antimicrobial activity was determined for Staphylococcus aureus; Escherichia coli and Candida albicans.
Overall, the study is well-designed and well-written, the methods are suitable and the obtained results are clearly presented and discussed. Moreover, the conclusions have been appropriately pointed out.
The content of this manuscript matches well with the journal’s purpose, but some clarifications are needed:
line 147: should be "...40°C. 100 μL of Tween 80 (T80) emulsifying agent was added at room temperature, while 900 μL of MH..."
line 191: "p" should be italic
line 251: Busatta et al. (2007) ?? Change it.
line 254: should be "...case of S. aureus..."
line 257: Sarikurkcu et al. (2015)???
line 264: Ličina et al. (2013)???
line 272: Martino et al. (2009)???
line 282: Özkalp et al. (2010)???
The authors must decide that they use units as e.g. mg/g or mg g-1. It is important. you must correct all the papers.
Although the manuscript is well written, correct and unambiguous, I recommend a carefully check of the full manuscript to correct any grammatical or syntax errors.
Author Response
Reviewer 1
Dear Reviewer, we appreciate the time and efforts of you and the referees in reviewing our manuscript.
We believe Your comments have been satisfactorily addressed. All the changes are visible in track changes in the Manuscript.
This manuscript describes to determine the optimal extraction conditions by applying the Design-Expert pro 11 software, with time and temperature being the observed variables. Further, supercritical fluid extract was carried out, and the effects of pressure were also examined. In the first step, the extracts were evaluated for their antioxidant activity for a preliminary indication of their biological potential by using the DPPH method. Higher antioxidant activities were found with the supercritical fluid extraction (SFE) extracts at higher temperatures and pressures. Antimicrobial activity was determined for Staphylococcus aureus; Escherichia coli and Candida albicans. Overall, the study is well-designed and well-written, the methods are suitable and the obtained results are clearly presented and discussed. Moreover, the conclusions have been appropriately pointed out.
The content of this manuscript matches well with the journal’s purpose, but some clarifications are needed:
line 147: should be "...40°C. 100 μL of Tween 80 (T80) emulsifying agent was added at room temperature, while 900 μL of MH..."
Response: It has been corrected.
line 191: "p" should be italic
Response: It has been corrected.
line 251: Busatta et al. (2007) ?? Change it.
Response: References had been cited according to guidance for authors, but if it is needed, we will write it differently.
line 254: should be "...case of S. aureus..."
Response: It has been corrected.
line 257: Sarikurkcu et al. (2015)???
Response: References had been cited according to guidance for authors, but if it is needed, we will write it differently.
line 264: Ličina et al. (2013)???
Response: References had been cited according to guidance for authors, but if it is needed, we will write it differently.
line 272: Martino et al. (2009)???
Response: References had been cited according to guidance for authors, but if it is needed, we will write it differently.
line 282: Özkalp et al. (2010)???
Response: References had been cited according to guidance for authors, but if it is needed, we will write it differently.
The authors must decide that they use units as e.g. mg/g or mg g-1. It is important. you must correct all the papers.
Response: Units has been corrected into “mg/g” format.
Although the manuscript is well written, correct and unambiguous, I recommend a carefully check of the full manuscript to correct any grammatical or syntax errors.
Response: Manuscript has been checked and hopefully all mistakes have been corrected.
Reviewer 2 Report
This manuscript describes the optimization of a method for the extraction of Origanum vulgare by maceration and supercritical fluid extraction to obtain phenolic compounds with antimicrobial and antioxidant properties.
I have some comments for the paper, which must be taken into account to improve the article before it will be suitable for acceptance for publication.
- English (writing): the work should be checked once again carefully.
- In the title, authors should include the name of the specie in italics “Origanun vulgare”, as well as the word “in vitro” (line 173).
- Both in the abstract and in the conclusions a comparison was made with the Clevenger and Soxhlet techniques, but data is lacking in the discussion regarding to Soxhlet.
- No results should be displayed in the introduction section. For example, delete the phrase “Higher antioxidant activities were found with the supercritical fluid extraction (SFE) extracts at higher temperatures and pressures.”
- During the maceration process, please indicate the next step after removing the solvent with the rotary evaporator.
- Please, indicate information on the centrifuge used and replace "rpm" of the centrifuge with "g" units (line 151).
- Figure 4, please indicate the meaning of “ME”.
- The phrase “The results obtained by SFE at the lowest temperatures are like those obtained by maceration.” does not agree with the results in Figure 4. At 60°C are similar, but at 40°C they do not seem similar.
- Please, clarify, if the determination of the MIC was made in 5 different extracts (line 238) or in 2 different extracts (line 246).
- In Table 1, special mention should be made of the fact that the results at 83ºC were obtained with the extracts from maceration extraction because it seems that they were all obtained with the SFE method.
- The phrase “The oregano extract obtained under a pressure of 25 MPa and temperature of 40°C had the most significant impact on each tested microorganism.” does not agree with the data presented in Table 1. For example, for S.aureus, the lowest MIC was obtained with the SFE conditions of 60°C and 25 MPa.
- Please, give more information in reference 21.
Author Response
Reviewer 2
Dear Reviewer, we appreciate the time and efforts of you and the referees in reviewing our manuscript.
All the changes are visible in track changes in the Manuscript. Your comments have been addressed – please see the response below:
This manuscript describes the optimization of a method for the extraction of Origanum vulgare by maceration and supercritical fluid extraction to obtain phenolic compounds with antimicrobial and antioxidant properties.
English (writing): the work should be checked once again carefully.
Response: The Manuscript has been checked and corrected.
In the title, authors should include the name of the specie in italics “Origanun vulgare”, as well as the word “in vitro” (line 173).
Response: Phrases have been changed into italic.
Both in the abstract and in the conclusions a comparison was made with the Clevenger and Soxhlet techniques, but data is lacking in the discussion regarding to Soxhlet.
Response: Discussion has been extended with comparison regarding the Soxhlet method, lines 264-268.
No results should be displayed in the introduction section. For example, delete the phrase “Higher antioxidant activities were found with the supercritical fluid extraction (SFE) extracts at higher temperatures and pressures.”
Response: It has been removed from the Introduction part.
During the maceration process, please indicate the next step after removing the solvent with the rotary evaporator.
Response: It has been added, lines xx-xx.
Please, indicate information on the centrifuge used and replace "rpm" of the centrifuge with "g" units (line 151).
Response: It has been converted (25000 rpm is 70000 g). It was corrected in the Manuscript.
Figure 4, please indicate the meaning of “ME”.
Response: It has been added.
The phrase “The results obtained by SFE at the lowest temperatures are like those obtained by maceration.” does not agree with the results in Figure 4. At 60°C are similar, but at 40°C they do not seem similar.
Response: The sentences have been rephrased to be clearer (lines 221-223) and also number values have been added to the graph in Figure 4.
Please, clarify, if the determination of the MIC was made in 5 different extracts (line 238) or in 2 different extracts (line 246).
Response: It has been corrected in line 246, the MIC determination of five different extracts had been obtained.
In Table 1, special mention should be made of the fact that the results at 83ºC were obtained with the extracts from maceration extraction because it seems that they were all obtained with the SFE method.
Response: One column has been added in Table 1 to clarify.
The phrase “The oregano extract obtained under a pressure of 25 MPa and temperature of 40°C had the most significant impact on each tested microorganism.” does not agree with the data presented in Table 1. For example, for S.aureus, the lowest MIC was obtained with the SFE conditions of 60°C and 25 MPa. - se slišimo - ni jasno kaj želi
Response: Mentioned conditions have been chosen as optimal for each three microorganisms, not individually.
Please, give more information in reference 21.
Response: The reference has been edited.